# Influence of Aerobic Pretreatment of Poultry Manure on the Biogas Production Process

**Mantas Rubežius \*, Kęstutis Venslauskas** **, Kęstutis Navickas and Rolandas Bleizgys**

Agriculture Academy, Vytautas Magnus University, K. Donelaičio g. 58, LT-44248 Kaunas, Lithuania;
kestutis.venslauskas@vdu.lt (K.V.); kestutis.navickas@vdu.lt (K.N.); rolandas.bleizgys@vdu.lt (R.B.)

\* Correspondence: mantas.rubezius@vdu.lt

**Abstract:** Anaerobic digestion of poultry manure is a potentially-sustainable means of stabilizing this waste while generating biogas. However, technical, and environmental protection challenges remain, including high concentrations of ammonia, low C/N ratios, limited digestibility of bedding, and questions about transformation of nutrients during digestion. This study evaluated the effect of primary biological treatment of poultry manure on the biogas production process and reduction of ammonia emissions. Biogas yield from organic matter content in the aerobic pretreatment groups was 13.96% higher than that of the control group. Biogas production analysis showed that aerobic pretreatment of poultry manure has a positive effect on biogas composition; methane concentration increases by 6.94–7.97% after pretreatment. In comparison with the control group, $NH_3$ emissions after aerobic pretreatment decreased from 3.37% (aerobic pretreatment without biological additives) to 33.89% (aerobic pretreatment with biological additives), depending on treatment method.

**Keywords:** anaerobic digestion; ammonia mitigation; aerobic pretreatment; poultry manure management

---

## 1. Introduction

The poultry industry is growing rapidly along with the human population, which results in large quantities of animal wastes to be treated. The data shows that, on average, there are approx. $1.886 \times 10^9$ poultry heads in Europe and this generates more than $10^7$ tonnes of poultry manure, which is commonly used as a fertilizer in agriculture [1–3]. Improper handling of poultry manure can cause significant environmental damage. Ammonia ($NH_3$) and greenhouse gases (GHG) can pollute the air and improper use of nitrogen and phosphorus can lead to eutrophication and pollution of surface and groundwater and soil [4,5]. $NH_3$ and nitrogen oxide ($N_2O$) emission is involved in the formation of particulates, which pose a significant risk to human health [5–7]. Therefore, key emissions from poultry farms such as $N_2O$, methane ($CH_4$), hydrogen sulfide ($H_2S$), carbon dioxide ($CO_2$), $NH_3$, and particulate matter should be controlled.

Anaerobic digestion (AD) is a well-known technology that is widely used to stabilize organic waste. Anaerobic processes produce energy-rich biogas and digestate that can be used in agriculture as valuable fertilizers [8]. Therefore, the fermentation of poultry manure is considered an appropriate option for the management of such waste. However, the high organic nitrogen content (30 g/kg fresh weight) and low C/N ratio (7:1) is a major drawback that inhibits the anaerobic process. Anaerobic digestion of uric acid and undigested proteins results in high levels of unbound ammonia and ammonium ions. In such cases, even at low organic load, the ammonia concentration exceeds the inhibition threshold and causes process interruption in poultry manure processing biogas plants [4,8–10].

Research studies have been conducted on various physical, chemical, and biological methods that aim to reduce the inhibitory effects of ammonia and increase methane yield [11–14]. Such techniques



include optimization of the C/N ratio or trace elements, pH or temperature control, high volume dilution of water, use of zeolite and biochar, and contact membrane technology [12,15–17]. These mentioned practices quickly provide optimal biogas production, but they can be difficult to implement economically and strategically, and the environmental aspect remains unclear [11,18–22]. Biological methods can be a solution to these issues. Proper microbial and enzyme sources can increase the degradation of the raw material, increase methane yield, and shorten composting time. Biological pretreatment of feedstock can be a cost-effective and environmentally-friendly method to optimize the biogas production process and reduce ammonia inhibition by treatment waste such as poultry manure.

Thus, the present work aims to evaluate the influence of aerobic pretreatment of poultry manure on biogas production process potential and on ammonia emission. This study was performed in three steps—aerobic (biological) pretreatment, biogas production, and ammonia emission measurements. The aerobic pretreatment was carried out in a bioreactor with controlled environment conditions and inoculation by selected biological additives (microorganisms and enzymes) for the treatment of poultry manure. In the second stage the influence of pretreatment on the biogas production process was evaluated. In the third stage, direct ammonia emission measurements were performed using a laser spectroscopy analyzer. Ammonia emissions were measured from untreated poultry manure, poultry manure after pretreatment with biological methods, and anaerobic digestates.

## 2. Methodology

### 2.1. Raw Material Preparation

Poultry manure was taken from a broiler farm in southwestern Lithuania, where chickens are kept on peat litter. Manure was harvested in September 2019. The poultry manure was chopped and sieved through a 0.5 cm sieve to remove large particles and non-biodegradable material. The manure was then packaged in an airtight containers of 20 L and stored at 5 °C until use. Manure was diluted with tap water up to 85% moisture (15% dry matter) before use.

### 2.2. Pretreatment of Poultry Manure

Aerobic pretreatment of poultry manure was performed in a 5 L vertical cylindrical digester (Figure 1, aerobic pretreatment). A temperature of 30 °C was maintained using thermostatic control (10) and an electric heating pad (8). The substrate was agitated with a mixer (2) at 10 rpm. The substrate was aerated 10 times daily for 15 min at 1 L/min using a HAILEA ACO-9602 air blower (5) and a ceramic 107 mm diffuser (7) to maintain aerobic conditions. The amount of gas formed was measured with RITTER MilliGas counters (11). The gas collected was analyzed with an ETG Portable Biogas Analyzer mod. MCA BIO-P ($CH_4$ 0–100%, ±0.2%; $CO_2$ 0–50%, ±0.2%; $O_2$ 0–25%; $H_2S$ 0–10,000 ppm; and $H_2O$–40,000 ppm; CO 0–1%).

In order to improve the abundance and operational efficiency of the microorganisms, biological additives were inoculated for the pretreatment of the raw material. Biological additive is an industrially-prepared powdered biological additive. The supplement contains natural microorganisms that have been specifically selected for their ability to efficiently mobilize manure, compete successfully with anaerobic microorganisms to prevent the formation of gases such as $H_2S$ and $CH_4$, and effectively remove $NH_3$ and biodegradable organic matter. This biological supplement included *Bacillus subtilis*, *Bacillus megaterium*, *Bacillus licheniformis*, *Bacillus amyloliquefaciens*, *Bacillus thuringiensis*, cellulase, corn, wheat bran, and Yucca extract.

Raw poultry manure and substrates obtained with different pretreatment biological methods were named, respectively, as RPM, E0, and E1 (Table 1).

The biological additive was in dry form, so rehydration was required before use—10 g of the additive was mixed well with 100 mL tap water (~30 °C) and left to stand for 1 h. Then, 10 mL of the reconstituted solution was mixed with 5 L to 15% total solids (TS) diluted poultry manure (1.4 L raw poultry manure + 3.6 L tap water).

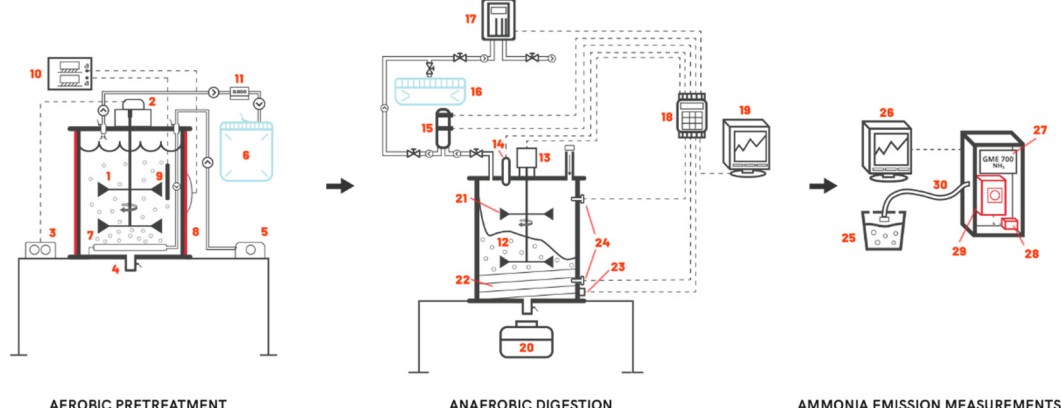

**Figure 1.** Research scheme: 1, pretreatment digester; 2, agitator drive; 3, agitator controller; 4, substrate drain valve; 5, blower; 6, gas bag; 7, ceramic diffuser; 8, electric heating pad; 9, temperature sensor; 10, heating controller; 11, milligas counter; 12, laboratory anaerobic digester; 13, gear of mixer; 14, pH electrode; 15, biogas flowmeter; 16, gas bag; 17, biogas analyzer; 18, programmable logic controller; 19, computer; 20, digested substrate vessel; 21, mixer; 22, heater; 23, relay of heating system; 24, temperature sensors; 25, poultry manure substrate; 26, computer; 27, GME700 laser gas analyzer; 28, membrane air pump; 29, electrically-heated three-channel valves; 30-heated air supply hose.

**Table 1.** Preprocessing scenarios.

| Experiment Code | Method of Treatment | Duration of Processing, Days |
|---|---|---|
| RPM | Pretreatment was not applied | |
| E0 | Maintenance of aerobic conditions without inoculation of biological additives | 14 |
| E1 | *Bacillus subtilis*, *Bacillus megaterium*, *Bacillus licheniformis*, *Bacillus amyloliquefaciens*, *Bacillus thuringiensis*, cellulase, corn, wheat bran, and Yucca extract. | 14 |

*2.3. Anaerobic Digestion Process*

Anaerobic digestion of poultry manure substrates was carried out in a cylindrical continuous-operation laboratory biogas digester (Figure 1 anaerobic digestion), consisting of a 5 L volume glass fiber vessel with substrate mixer (22 rpm for 105 s). The digester was maintained in a mesophilic environment at $37 \pm 1\,^\circ$C temperature and volumetric organic loading rate of 1 kg m$^3$/d (calculated by RPM). Temperature, alkalinity of the substrate, and biogas yield data were recorded by a programmable logic controller and stored in the computer database. The produced biogas was collected through the volumetric drum-type biogas flowmeter in a gasholder (25 L RESTEK 22967 bag, Bellefonte, United States). The collected biogas was analyzed by a biogas analyzer Geotech Biogas 5000, Warwickshire, United Kingdom. The biogas production process lasted 35 days.

The results of biomass anaerobic digestion were evaluated using the following indicators: intensity of biogas production, biogas yield from fresh biomass ($B_M$), biogas yield from biomass total solids ($B_{TS}$), biogas yield from biomass volatile solids ($B_{VS}$), and energy value of biomass obtained at anaerobic conversion ($e_M$, $e_{TS}$, and $e_{VS}$). The intensity of biogas production (b) indicated the duration of biomass biological degradation. The biogas yield from the biomass, from biomass total solids, and from biomass volatile solids $B_M$, $B_{TS}$, and $B_{VS}$ was calculated by the equations [23]: $B_M = b_{dt}/m$; $B_{TS} = b_{dt}/m_{TS}$; $B_{VS} = b_{dt}/m_{VS}$, where $b_{dt}$ is the volume of produced biogas during the time interval dt, L; m is the mass of the sample, kg; $m_{TS}$ is the mass of total solids of the sample, kg; and $m_{VS}$ is the mass of volatile solids of the sample, kg.

The energy value of the biomass obtained at anaerobic digestion ($e_M$, $e_{TS}$, and $e_{VS}$) is determined by the equations [23]: $e_M = B_M \cdot e_b$; $e_{TS} = B_{TS} \cdot e_b$; $e_{VS} = B_{VS} \cdot e_b$, where $e_b$ is the energy value of biogas

which depends on the methane concentration in the biogas, MJ/L. The energy value of the biogas is determined by the equation [23]: $e_b = 0.0353 \cdot C_{CH4}/100$, where $C_{CH4}$ is the methane concentration in biogas, %.

### 2.4. Ammonia Emission Measurements

Ammonia emissions were measured from untreated poultry manure, poultry manure after pre-treatment with biological methods, and anaerobic digestates. The GME700 (SICK MAIHAK GmbH, Meersburg, Germany) analyzer was used to measure ammonia gas concentration by laser spectroscopy in the measurement range for $NH_3$ from 0 ppm to 2000 ppm (Figure 1) [24].

The sample taken was placed in a plastic 1 L container with a surface area of 0.0095 $m^2$. Air was supplied to the analyzer by a continuous pump with a flow rate of 6 L/min. To prevent contamination of the cell and condensation, the suction gas was heated in the suction hose to 150 °C in electrically-heated valves.

### 2.5. Chemical Analysis

The chemical composition was been determined by an accredited Agrochemical Research Laboratory. The pH was measured by HANNA instruments pH 213 Microprocessor pH Meter.

### 2.6. Data Processing

Each experiment was performed in triplicate. Statistica 10 software (StatSoft, Hamburg, Germany) was used to process information. The mean of the test results was compared using the t (*t*-test) criterion. Significant differences were reported at a *p*-value lower than 0.05.

## 3. Results and Discussion

### 3.1. Aerobic Pretreatment

### 3.1.1. Chemical and Physical Properties

The poultry manure after 14 days of aerobic pretreatment loses some of the organic carbon that is converted into $CO_2$ and other inorganic materials during the process [18]. Therefore, the total carbon content was reduced in all experimental groups. The highest total carbon loss was found for E1. This significant reduction (2.01%) in total organic carbon was influenced by the highly-efficient metabolic activity of the inoculated microorganisms (Table 2).

**Table 2.** Results of chemical analysis of substrates.

| Research Parameter | Research Code | | | | | |
| --- | --- | --- | --- | --- | --- | --- |
| | Raw Material | | | Digestate | | |
| | RPM | E0 | E1 | RPM | E0 | E1 |
| Dry matter (TS), % | 15.00 | 14.47 | 12.96 | 8.94 | 9.62 | 6.49 |
| Organic matter (VS), % | 12.83 | 10.49 | 11.18 | 5.51 | 6.94 | 4.58 |
| Total nitrogen (TN), % | 0.85 | 0.86 | 0.92 | 0.78 | 0.85 | 0.66 |
| Sulfur (S), mg/kg | 1246 | 1120 | 1059 | 387 | 557 | 373 |
| Ammoniacal nitrogen (TAN), g/L | 2.13 | 4.80 | 5.17 | 5.63 | 4.82 | 4.40 |
| Total carbon (TC), % | 7.33 | 6.75 | 5.32 | 4.06 | 4.46 | 3.07 |
| $NO_3^-$, mg/L | <2.00 | <2.00 | <2.00 | <2.00 | <2.00 | <2.00 |
| $NO_2^-$, mg/L | <0.05 | <0.05 | <0.05 | <0.05 | <0.05 | <0.05 |
| pH | 5.65 | 6.17 | 6.25 | 7.88 | 7.91 | 7.66 |
| C/N | 9/1 | 8/1 | 6/1 | 5/1 | 5/1 | 5/1 |

$NO_2^-$ and $NO_3^-$ studies were conducted to determine the extent of ammonia oxidation (nitrification) during biological treatment of poultry manure and to determine the potential effect of

oxidation on anaerobic digestion of the treated substrate. According Li et al. [25] the peak period of ammonia emissions is between 10 and 30 days, and the time when the total nitrogen (TN) content drops sharply is also in this period. In a later phase of composting (maturation), nitrifying bacteria begin to activate, which results in a rapid decrease in $NH_3$ concentration. In this study, the pretreatment was carried out at 30 °C, which, according to Caceres et al. [26], is the optimum temperature for nitrification (20–35 °C). However, differences in $NO_3^-$ between treated and untreated poultry manure were not found. In addition, many contributions state that high ammonia content and low C/N ratios (optimal nitrification C/N is 28) can limit nitrification [26–31].

After the primary treatment of poultry manure, moisture increased. Such increases are commonly observed in closed treatment systems, where metabolic water produced by microbial activity exceeds moisture losses by evaporation [18,32]. TS losses were estimated to increase from 0.53 (E0) to 2.04% (E1) in the TS-treated substrate after pretreatment (Table 2). The volatile solid (VS) losses were estimated to decrease from 1.65% (E1) to 2.34% (E0) in the VS-treated substrate after pretreatment (Table 2). Changes in dry organic matter can be seen as results of more-active microbial activity when some organic matter is converted to gases such as $CO_2$, $H_2$, $H_2S$ and $NH_3$.

### 3.1.2. Gas Emission by Aerobic Pretreatment

One of the disadvantages of the aerobic pretreatment method is that this method is susceptible to the time required for incubation of microorganisms. The yield of gas was recorded at the same time each day. Judging from the intensity of gas formation (Figure 2), the most intensive microbiological activity took place in the first days of aerobic treatment, in the RPM case on days 2–3, in the E0 case, on days 1–3, and in the E1 case on days 1–3. Thereafter, the activity gradually decreased and stabilized at day 7. These gas formation intensity results correlated well with the pH change results. During the first 7 days of treatment, the pH rose and then stabilized. The final pH values of E0 and E1 were 6.34 and 6.28, respectively. In the case of RPM, the pH remained almost unchanged at 5.47. pH changes are associated with the consumption of organic acids, $CO_2$ evaporation, and the accumulation of $NH_4^+$-N [29,33–35]. After evaluating the obtained results, we can assume that we can reduce the duration of the selected treatment by half (up to 7 days). Optimal processing times can reduce costs and increase the attractiveness of the method, but additional studies are needed to confirm this assumption.

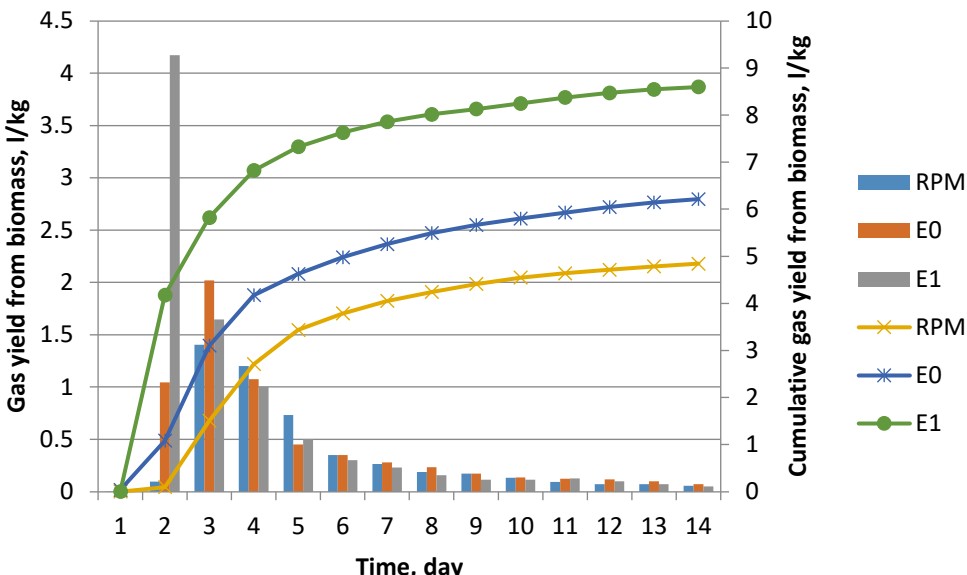

**Figure 2.** Amount of gas formed during pretreatment.

Comparing the study groups, it was found that most of the gas was formed in the case of E1, when favorable conditions for hydrolysis were maintained and the additive of hydrolytic microorganisms was used. Results differed more than twice between E1 and other study groups.

The gases formed during the pretreatment were continuously collected. The gas composition analysis was divided into two periods, days 1–7 and days 7–14. The gases emitted during the pretreatment process consisted mainly of $CO_2$, $H_2$, and $H_2S$, the ratio of which changed over time (Table 3). Other analyzed gases such as $CH_4$, $O_2$, CO, and $NH_3$ were not detected or their values were low. On days 1–7, the $CO_2$ concentration in the case of RPM was 52.06%, in the case of E0 it was 49.13%, and in the case of E1, 48.90%. The $H_2S$ concentration in the case of RPM was 510 ppm, for E0 it was 5766 ppm, and for E1, 2479 ppm. The $H_2$ concentration in the case of RPM was 21,530 ppm, for E0, 21,450 ppm, and for E1, 21,051 ppm.

**Table 3.** Gas composition after aerobic pretreatment.

|  |  | $CH_4$, % | $CO_2$, % | CO, % | $O_2$, % | $H_2S$, ppm | $H_2$, ppm | $NH_3$, ppm |
|---|---|---|---|---|---|---|---|---|
| | RPM | 1.53 | 52.06 | 0.62 | 4.27 | 510 | 21,530 | 3.27 |
| Days 1–7 | E0 | 0.00 | 49.13 | 0.02 | 1.50 | 5766 | 21,450 | 4.00 |
| | E1 | 0.12 | 48.90 | 0.00 | 5.72 | 2479 | 21,051 | 3.57 |
| | RPM | 0.40 | 39.20 | 0.01 | 13.34 | 0 | 399 | 2.24 |
| Days 7–14 | E0 | 0.27 | 45.35 | 0.00 | 5.83 | 114 | 2049 | 2.90 |
| | E1 | 0.11 | 27.53 | 0.00 | 1.94 | 17 | 0 | 3.57 |

Interestingly, the concentration of ammonia in the gas generated during the treatment of poultry manure in a closed vessel was only at the background level (> 4 ppm). Most of the ammonia formed remained in $NH_4^+$-N form. Compared to RPM, E0, and E1 TAN content in biomass increased by 2.97 and 3.72 g/L, respectively (Table 2). However, the increase in TAN content during treatment did not affect the change in $NH_3$. For all cases, only traces of $NH_3$ were found in the gas, which were between 3 and 5 ppm. Some of the $NH_3$ formed could have been absorbed on the surface of the reactor due to the water vapor layer formed. The formed water condensate returns back to the biomass while returning the dissolved ammonia back [25]. However, when conditions change, exposure to ambient air changes the situation radically and ammonia emissions increase sharply (Figure 3).

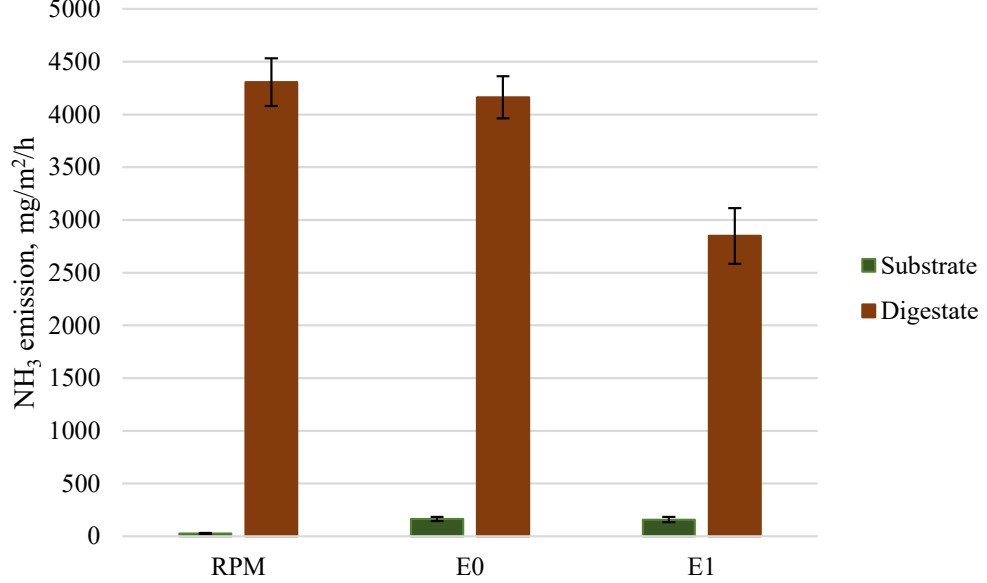

**Figure 3.** Average $NH_3$ emission from substrate per 24 h.

### 3.2. Biogas, Methane Yields, and Energy Value

Table 4 presents the results of 35 days anaerobic digestion of the biomass of the different poultry manure substrates. The lowest biogas yield from fresh matter (M) was found in the E0 group (53.44 L/kg) and the highest in the E1 group (56.65 L/kg). Statistical analysis of the data showed that there was a statistically-significant difference ($p > 0.05$). Interestingly, the biogas yield was not proportional to the TS loading rate.

**Table 4.** Biogas, methane yields, and energy value.

| Indicator. | RPM | E0 | E1 |
|---|---|---|---|
| Biogas yield from biomass ($B_M$), L/kg | 56.38 ± 3.19 | 53.44 ± 3.16 | 56.65 ± 2.20 |
| Biogas yield from dry matter ($B_{TS}$), L/kg | 375.87 ± 21.30 | 403.95 ± 23.95 | 422.72 ± 16.51 |
| Biogas yield from organic matter ($B_{VS}$), L/kg | 439.45 ± 24.90 | 510.78 ± 23.86 | 507.98 ± 19.65 |
| Methane concentration in biogas ($C_M$), % | 54.92 ± 1.98 | 61.86 ± 1.39 | 62.89 ± 1.15 |
| Energetic value of biogas ($e_b$), MJ/m$^3$ | 19.39 ± 0.69 | 21.84 ± 0.49 | 22.20 ± 0.41 |
| Energy obtained from biomass ($e_M$), MJ/kg | 1.09 ± 0.06 | 1.31 ± 0.08 | 1.26 ± 0.05 |
| Energy obtained from dry matter ($e_{TS}$), MJ/kg | 7.28 ± 0.42 | 8.82 ± 0.54 | 9.37 ± 0.41 |
| Energy obtained from organic matter ($e_{VS}$), MJ/kg | 8.52 ± 0.49 | 11.04 ± 0.10 | 11.27 ± 0.49 |

TS, total solids; M, fresh matter; VS, volatile solids.

Bremond et al. [36] have comprehensively reviewed and summarized the effect of each pretreatment on methane yield with respect to the feedstock. Depending on the characteristics of the raw materials, their effects can vary widely, from positive to negative effects on the subsequent anaerobic process. Despite the greater degradation of lignocellulose, VS loss is not taken into account when evaluating the beneficial effects, and therefore many studies are unclear or actually show improved methane yield [36]. In this study, VS measurements revealed (Table 2) that the greatest change in VS after pretreatment was for E0 (difference 2.34%). However, according to the AD process assay, VS losses after pretreatment did not adversely affect methane yield. In addition, VS loss is not considered a problem when it comes to reducing waste amount.

Recalculation of biogas yield from dry organic matter revealed that in this case the highest biogas yield was found in experimental group E0 (510.78 L/kg VS) and the lowest in the RPM group (439.45 L/kg VS) (Table 4). However, there was no statistically-significant difference ($p > 0.05$) between the E0 and E1 groups. The analysis of biogas quality revealed that the highest methane concentration was recorded in the case of E1 (62.89%) and the lowest in RPM (54.92%) (Table 4). Statistical analysis of the data showed that there was a statistically-significant difference ($p > 0.05$). Overall, for all experimental groups, any pretreatment method for poultry manure had a positive effect on methane concentration, with differences ranging from 6.82% to 7.96%.

The averaged data of energy values (Table 4) showed that the best energy value from biomass ($e_M$) was found in the E0 group (1.31 MJ/kg) and the worst in the RPM group (1.09 MJ/kg). When energy value results from organic matter ($e_{VS}$) were evaluated, it was found that the best results were obtained in E1 group (11.27 MJ/kg VS) and the worst in RPM group (8.52 MJ/kg VS). A longer duration of pretreatment is closely associated with higher energy consumption and may be unprofitable in the end result. So further studies and calculation are needed to confirm this.

According to previous research, pretreatment of the substrate optimizes the degradation process, degradation intensity, release of intracellular nutrients, and increases methane yield [18,22,37,38]. These enhancements can be quantified by higher VS degradation, faster biogas and methane production rates and ultimate yields, and lower volumes of residual solids for final disposal after the AD process.

Pretreatment of biomass also alters the ratio and availability of other components such as organic acids, nitrogenous substances, and free sugars. The anaerobic process with biomass pretreatment avoids the inhibitory effect on acetoclastic methanogens resulting from the accumulation of the volatile fatty acids (VFA) [39]. Pretreatment intensifies biological processes such as hydrolysis and ammonification

that result in altered C/N ratios (Table 2). Part of the organic carbon is lost in the form of $CO_2$, and more nitrogen (N) is liberated as a result of intensified nitrogen mineralization [32]. Wang et al. [40] found that the accumulation of $NH_4^+$ at low C/N ratios creates suitable conditions for methanogens, since the volatile fatty acids, VFA and $NH_4^+$, formed during acetogenesis formed a methanogen buffer system, resulting in improved methanogen reactivity and productivity, resulting in more VFAs being converted to methane.

After pretreatment, hydrogenotrophic methanogenesis is favored. Pretreatment readily converts acetate to hydrogen and carbon dioxide, which is converted to biogas by hydrotropic methanogens during the anaerobic process; consequently acetoclastic inhibition reduces, resulting in increased methane yield [41–43]. Acs et al. [38] found a positive correlation between biogas production and the strain of hydrogen-producing microorganism introduced into the biogas production process. Hydrogen-producing strains increase the amount of hydrogen, which in turn increases methane production, reduces $CO_2$, and increases the purity of the biogas produced. Wang et al. [44] found that bacteria sensitive to high ammonia levels could tolerate higher ammonia levels when hydrogen is added. In addition, the present study indicated that hydrogen injection under high ammonia concentration might also strengthen the growth of some hydrolytic and fermentative bacteria during the biogas production process.

In the E1 case, optimization of the biodegradation process and higher methane concentration were influenced by inoculation of selected microorganisms and enzymes. Strains of microorganisms such as *Bacillus spp.* have great potential for lignocellulosic processing to promote biogas production [21]. As a microbial supplement, enzymes can provide optimal growth and activity for various types of microorganisms, making biomass more resistant to shock load. Enzymes are also used to eliminate the disadvantages associated with the use of conventional chemical catalysts [18,36,45].

### 3.3. Ammonia Emission

Emission of ammonia from substrate and digestate is reported in Figure 3. Ammonia concentration was recorded continuously every 1 min, the studies lasted 24 h, the total score was more than 1440 points. Ammonia emissions were found to be higher than those from untreated poultry manure in all cases of pretreatment of litter. The highest daily evaporation of ammonia from the substrate was observed for E0 (162 mg/m$^2$/h) and the lowest for RPM (25 mg/m$^2$/h). There was a statistically-significant difference between the means of all variants at a 95% confidence level.

Pretreatment activates a variety of biological processes, including ammonification, the biochemical release of ammonia from nitrogenous organic materials. Table 2 clearly shows that $NH_4^+$-N increases more than 2-fold after pretreatment in all experimental cases. Changes in pH and VS are also important factors in increasing ammonia emission. The increase in pH value in biological pretreatment was mainly the result of ammonification [25]. An increase in pH contributes to the disturbance of the ammonia balance, beginning with the conversion of ionized $NH_4^+$, which is not volatile, to $NH_3$, with lower solubility and higher evaporation [30,46,47]. Slurry and digestate residues with higher solids content have lower nitrogen removal efficiency than those with low solids content, likely due to the binding of ammonium ions to organic matter [48].

The ammonia emissions from the digestate after the AD process showed that the highest $NH_3$ emissions from the digestate were found for RPM (4308 mg/m$^2$/h) and the lowest for E1 (2848 mg/m$^2$/h). There was a statistically-significant difference between the means of all variants (Figure 3). Unfortunately, from an environmental point of view, the maximum evaporative values for $NH_3$ from waste facilities are not precisely defined. However, the results obtained confirm the need to pay due attention to the reduction of $NH_3$ emissions at both local and regional levels.

An optimum ammonia concentration ensures adequate buffering capacity, however, it may also lead to inhibition of anaerobic digestion at high concentrations. Generally, inhibitory TAN concentrations are reported in the range of 3–6 g/L [49]. Table 2 shows that the mean TAN concentration in all assays increased after aerobic pretreated. In any case, the concentration of TAN after AD

did exceed 2 g/L, thereby suggesting that inhibitory TAN concentration was present in these assays. However, in E1 case where biological additives were used the TAN concentration was lower than RPM and E0.

Based on the results of the ammonia emission (Figure 3) and biomass conversion to biogas (Table 4), it can be assumed that pretreatment of poultry manure increases the activity, stability, and tolerance of anaerobic microorganisms to higher concentrations of ammonia. By increasing the activity and tolerance of microorganisms, more $NH_4^+$ is immobilized via microbial metabolism, leading to lower $NH_3$ emissions [22,29,50]. Studies have shown that the $NH_4^+$-N consumed for the growth of hydrolytic bacteria and methanogens inoculated during the AD process exceeded its production through degradable proteins and amino acids, leading to a reduction in the ammonia nitrogen concentration. We can state that in the experimental groups the consumption rate of $NH_4^+$-N during the biogas production process was faster than in the control group (RPM).

On the other hand, after the AD process, a significant reduction in TN and TAN levels was observed when comparing treated and untreated poultry manure digestates. For E1, a decrease of 0.12% and 1.23 g/L in TN and TAN, respectively. Part of the TAN evaporates in the form of $NH_3$ as a component of biogas. After measuring the concentration of $NH_3$ in the formed biogas, it was found that the RPM cases in the total composition of $NH_3$ in biogas was 28.80 ppm, in the case of E0 it was 33.16 ppm, and in the case of E1, 37.44 ppm. Also, such losses make one think about the loss of $N_2$ in forms other than $N_2O$. It is likely that after pretreatment of poultry manure, stimulating conditions are created for the growth of ammonium ($NH_4^+$)-oxidizing bacteria (AOB), resulting in increased $N_2$ or $N_2O$ emissions [50,51]. However, further studies are needed to confirm this.

## 4. Conclusions

The present work aimed to evaluate the influence of aerobic pretreatment of poultry manure on the biogas production process and ammonia emission. Based on the results of biogas production, aerobic pretreatment of poultry manure has a positive impact on biogas yield from organic matter, biogas quality, and pretreated substrate degradation rate. Maintaining optimal conditions (heating, aeration, and mixing) and using biological additives during pretreatment increases the biogas yield from 68.53 to 71.33 L/kg organic matter. The concentration of methane in biogas after pretreatment was influenced by aerobic process and increased from 54.92% to 62.89%. The results obtained can be considered as the result of more efficient hydrolysis where the pretreatment of the substrate optimizes the biodegradation process and the intensity of degradation. Energy obtained from organic matter after pretreatment increased by 2.52–2.75 MJ/kg showing the impact of pretreatment.

Pretreatment of poultry manure has been shown to activate the ammonification process in substrate $NH_3$ emission research. In all pretreatments, substrate $NH_4^+$-N increases more than 2-fold. Ammonia emissions were also found to be 133–137 mg/m$^2$/h higher in all cases of litter treatment than untreated poultry manure. Measurements of ammonia emissions from the digestate after the anaerobic digestion process showed obvious differences between the test groups. $NH_3$ emissions after aerobic pretreatment without biological additives decreased to 145 mg/m$^2$/h. $NH_3$ emissions after aerobic pretreatment with biological additives decreased to 1460 mg/m$^2$/h.

Further studies are needed to confirm the benefits of aerobic pretreatment of poultry manure for the biogas production process. Not only the technical but also the economic feasibility of the biogas production process must be taken into consideration. Studies have shown that it is possible to reduce by more than twice, the ammonia emissions from anaerobic digestate. However, considering the nitrogen losses, there remains the unresolved question of nitrogen losses in other forms, such nitrous oxide.

**Author Contributions:** Resources, R.B.; data curation, K.V.; writing—original draft preparation, M.R.; writing—review and editing, K.N. All authors have read and agreed to the published version of the manuscript.

**Funding:** This research received no external funding.

**Conflicts of Interest:** The authors declare no conflict of interest.

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
