# Peer review of "Influence of Aerobic Pretreatment of Poultry Manure on the Biogas Production Process"

_processes, doi:10.3390/pr8091109_

Round 1
Reviewer 1 Report
This study investigated how to improve methane production and reduce ammonia by pre-treatment of poultry manure. The study objective and method are well introduced. The following suggestions are for authors’ consideration:
- Author affiliation is not following format of the journal.
- Citation format and reference list are not correct.
- Poultry manure/litter is relatively dry, which is different from dairy or swine production system where the manure contains high moisture. Therefore, using aerobic methods require adding water to the manure, which may trigger future questions: (1) liquid waste handling and water treatment; and (2) nutrient loss from crop production. Authors may address above concerns by discussing why energy is more important in some areas and how the liquid waste may be handed for location poultry farm who may interest to use aerobic digestion method. A suggestion would be highlighting the potential emissions of bacteria from poultry production and indicate that aerobic digestion may improve the situation. Some citations about poultry production bacteria emissions are listed for authors consideration for improving introduction or discussion of this writing:
Nimmermark, S., Lund, V., Gustafsson, G., & Eduard, W. 2009. Ammonia, dust and bacteria in welfare-oriented systems for laying hens. Annals of Agricultural and Environmental Medicine, 16(1), 103-113.
Zucker, B. A., Trojan, S., & Müller, W. (2000). Airborne Gram‐Negative Bacterial Flora in Animal Houses. Zoonoses and Public Health, 47(1), 37-46.
Chai, L., Zhao, Y., Xin, H., Wang, T., & Soupir, M. L. (2018). Mitigating airborne bacteria generations from cage-free layer litter by spraying acidic electrolysed water. Biosystems Engineering, 170, 61-71.
- Using additive is an important treatment for this study. The details of additive (e.g. chemical formula or bacteria name) should be included in Table 1.
- Line 144: The company name of Statistical software should be listed.
- title of table 2 could be more specific
- formula of gas need to be corrected by adjusting number as subscript.
- Line 212-214: How did you know most of ammonia are in molecular form? Did you test the percentage of ammonia in ammonium and organic N?
Reviewer 2 Report
The authors tested the effects of an aerobic pretreatment (composting) on the anaerobic digestion of poultry manure in laboratory scale. One pretreatment was done with a commercial additive and the other without. Tested were both variants of the pretreatment and the untreated control. Both pretreatment variants resulted in higher methane yields during the anaerobic digestion, without significant differences between them. The digestate that originated from the pretreatment with the additive showed significant lower ammonia emissions than the other variant or the control.
Dear authors,
Thank you very much for your interesting manuscript. After some general remarks you find specific points with questions and suggestions in order to improve your text, each referring to the respective line number.
A thing I missed in the introduction was the severity of the problem you work on. Could you provide some numbers regarding the amount of poultry manure (treated, untreated, emissions) in Lithuania, the EU, or the world?
In the Methodology I missed the information about the number of
iterations. Were the tests done in triplicates?
In the Results and discussion I expected some remarks in regard to the transferability of your results to large-scale operations. This could also have ignited some discussion about the costs and benefits in comparison to other pretreatment options.
I noticed that you often refer to previous research in order to explain your findings. This is of course correct, but you phrase it as facts, while mostly it just provides one possible explanation among others.
Abstract
017: Please define RPM or remove it.
1. Introduction
027: With „improper use of nitrogen an phosphorus“ you probably mean the pollution originating from the improper use of poultry manure.
029: Are all GHG emissions from poultry manure involved in the formation of particulates? If not, please specify.
031: How do you define „key poultry farms“?
046: With „Such techniques“ you seem to refer only to physical and chemical methods, since you later speak about biological methods. If you do not write that clearly it is hard to understand.
050: I can understand „difficult to implement economically“. Therefore you probably aim at a more cost efficient method. However, could you please specify the „strategically“ problems of the previous methods and their environmental aspect, which „remains unclear“?
054: With „shorten composting time“ you refer to a pretreatment method, before you introduced it. By the way, not only microbial and enzyme additives can speed up this process, but also C/N ratio optimization, or the addition of biochar (physical methods?).
2. Methodology
074: „Anaerobic“ shall probably be „Aerobic“.
083: Figure 1 provides a helpful overview about the research setup. Unfortunately the font and details are rather small.
092: Please declare in „conflicts of interest“ at the end of the paper that you either have no conflicts, or that there are connections between you and the producers or sellers of Additive B1.
096: When you „remove [...] biodegradable matter“, do you not diminish the input material for the AD process?
099: RPM (raw poultry manure) should not be addressed as a substrate „obtained with different pretreatment biological methods“, because it underwent no specific pretreatment.
101: Could you please clarify that E1 underwent the same pretreatment as E0, just with the additives?
103: You probably mean hydration instead of „dehydration“.
122: Why do you refer to a publication (Butkute et al. 2014), which was co-authored by some of the co-authors, for quite simple calculation methods? Did they not exist before?
129: Same question as for line 122.
135: Why do you refer to a publication (Bleizgys and Bagdoniene 2016), which was authored by a co-author, where simply the same analyzer was used?
136: Could you please explain, why the surface area of the container is relevant?
145: Is that correct: „Differences were considered statistically significant when significance was less than the significance level of 0.05.“?
3. Results and discussion
153: The „significant reduction“ probably have been influenced by the additives (correlation). However, you cannot write it as if it is proven, or is it?
162: You refer to results from composting research. However, your pretreatment method was done with a substrate that had a 85% moisture content, which is unusually high for composting.
164: The same comparison as in line 162, but to another publication.
170: How can there be moisture losses by evaporation in „closed“ composting systems?
172: Something is wrong with „decrease from 0.53 to 2.04%“. Same as in line 173.
196: It might not be the right section (pretreatment) to conclude that the additive pretreatment is „excellent“. In addition, the gas composition in Table 3 does not „unequivocally“ support your claim in regard to the pretreatment.
208: Your excursion into biological H2-production seems somewhat strange and is not supported by your data, or is it?
220: „...and ammonia emissions increase sharply.“ Is that by Li et al. (2020) or based on your own measurements?
235: Table 2 instead of „Table 4“?
241: Based on „no statistically significant difference (p> 0.05) between the E0 and E1“ one could conclude, that the additives were useless in regard to the biogas yield.
251: When RPM had the least energy value per organic matter content, why do you explain that with „E0 group had the lowest organic matter content“?
254: Should begin with „According to previous research...“
270: The whole paragraph is about previous research that has nothing to do with your data, or does it?
282: Your conclusion „In E1 case optimization of the biodegradation process and higher methane concentration were influenced by inoculation of selected microorganisms and enzymes.“ is contradicted by your results stated earlier (no significant differences between E0 and E1).
283: Your reference to previous research „Strains of microorganisms such as Bacillus sp. has great potential for lignocellulosic processing“ is of interest regarding poultry manure. However, how much lignocellulosic material did you have in your substrate after it went through a 5 mm sieve? In addition, your data does not corroborate this „great potential“, which would be a valid result as well.
288: Your reference to previous research might be true: „Preenzymatic hydrolysis can be an alternative to energy-intensive heat and mechanical treatment“. However, what has that to do with your research. E0 and E1 underwent the same treatment (temperature, aeration, mixing).
311: The highest emission from digestate according to Figure 3 is from RPM, not E0, is it not?
313: I do not think that the results of this study „revealed that average daily NH3 emissions from anaerobic digestion waste pose a significant risk to the environment and to biogas plant workers processing poultry manure“. You just confirmed previous research that there are relevant NH3 emissions.
Conclusions
351: Does your data really suggest that „using biological additives during pretreatment“ increases the biogas yield (significantly)?
362: Something wrong here: „...decreased from 145 to 1460 mg/m 2 /h“
363: Since AD is a biological treatment, the following phrase might not be ideal: „...biological treatment of poultry manure for
the biogas production process.“
Reviewer 3 Report
Manuscript ID: processes-898896
Title: Influence of aerobical pretreatment of poultry manure on biogas production process
Serious Concerns:
Abstract
In comparison with the control group, the NH3 emissions after aerobical pretreatment decreased from 3.37% to 33.89%, depending on treatment method. There is no information about used methods
Introduction
What is the novelty? What the paper is going to add to the existing knowledge?
Results and discussions:
It would be nice to see some more advanced methodologies in analysis of data presented in the text. Some data seem raw without more comprehensive elaboration
Conclusions
- A more precise discussion should be drawn. The Authors should explain exactly what they obtained. The paper should formulate clear and detailed objectives, clear and concise results, a logic interpretation and finally end with an enriching discussion and conclusion. Furthermore it brings nothing new and original which is worth publishing in the journal.
- Energy obtained from organic matter after pretreatment increases by 2.52-2.75 MJ/kg. Could authors please explain how this value was calculated?
Please pay attention to subscripts
On this basis, I recommend minor revision.
